# Electron affinity of liquid water

Alex P. Gaiduk [1], Tuan Anh Pham[2], Marco Govoni [1,3], Francesco Paesani [4] & Giulia Galli[1,3]

Understanding redox and photochemical reactions in aqueous environments requires a precise knowledge of the ionization potential and electron affinity of liquid water. The former has been measured, but not the latter. We predict the electron affinity of liquid water and of its surface from first principles, coupling path-integral molecular dynamics with ab initio potentials, and many-body perturbation theory. Our results for the surface (0.8 eV) agree well with recent pump-probe spectroscopy measurements on amorphous ice. Those for the bulk (0.1–0.3 eV) differ from several estimates adopted in the literature, which we critically revisit. We show that the ionization potential of the bulk and surface are almost identical; instead their electron affinities differ substantially, with the conduction band edge of the surface much deeper in energy than that of the bulk. We also discuss the significant impact of nuclear quantum effects on the fundamental gap and band edges of the liquid.

[1] Institute for Molecular Engineering, The University of Chicago, Chicago, IL 60637, USA. [2] Lawrence Livermore National Laboratory, Livermore, CA 94551, USA. [3] Materials Science Division, Argonne National Laboratory, Argonne, IL 60439, USA. [4] Department of Chemistry and Biochemistry, Materials Science and Engineering, San Diego Supercomputer Center, University of California, San Diego, 92093, USA. Correspondence and requests for materials should be addressed to F.P. (email: fpaesani@ucsd.edu) or to G.G. (email: gagalli@uchicago.edu)

Upon exposure to light or ionizing radiation, liquid water may undergo a series of transformations involving electronic excitations, ionization of solvated species, and formation of free radicals and solvated electrons[1–7]. These processes occur, for example, in (photo-)electrochemical cells[8,9], biological molecules[10], as well as in atmospheric water[11]. Many of them are governed, at least in part, by the nature of the electron transfer and binding energies in aqueous solutions. Hence a detailed understanding of the interaction of light with water, and in general of redox reactions in aqueous systems, requires a precise knowledge of the liquid ionization potential (IP) and electron affinity (EA).

Direct measurements of the IP of liquid water are available from photoelectron spectroscopy, obtained by linear extrapolation to zero energy of the lowest binding-energy peak ($1b_1$). The measurement of Delahay et al.[12] yielded 10.06 eV for the water IP, while more recent photoelectron spectroscopy studies led to a slightly smaller value of 9.9 eV[13], leading to an overall agreement that the water IP is ~10 eV. On the other hand, direct measurements of the liquid EA, or the energy gain due to the injection of an electron into the liquid, are not available, due to the rapid solvation of electrons in water[1,2,14]. Current estimates of the EA are based on thermodynamic arguments involving adiabatic and vertical ionization energies of the solvated electron[7,15,16], computed reorganization energies, and specific assumptions on the behavior of photoionized electrons in water. As a result, the reported values of EA span a wide range between 0 and 1.2 eV[7,15–19].

The only direct measurement of the EA of condensed aqueous systems was performed for the surface of a thin film of amorphous ice (a-ice) by pump-probe femtosecond spectroscopy[20,21]. Ice films only 2.5-bilayer thick were grown on a copper electrode, hence the measurement probed the energy of the lowest empty state of the aqueous surface. This study revealed a short-lived (~40 fs) spectral feature at −0.8 eV below the vacuum level, which was attributed to an electron residing at the bottom of the conduction band of amorphous ice interfaced with copper[20].

Multiple predictions of water IP and EA from quantum calculations[7,22–26] produced conflicting results depending on the structural models and electronic-structure methods employed. For example, many-body perturbation theory (MBPT) calculations using wavefunctions derived from density functional theory (DFT) and the PBE approximation[22] yielded a value of 0.7 eV for the EA (configurations were generated either with classical potentials or the PBE functional); this value appeared to be consistent with estimates reported in the literature[15]. The quasiparticle self-consistent GW calculations of Chen et al.[25], who generated a water structural model using van der Waals functionals and included nuclear quantum effects, obtained EA = 0.5 eV. A later thermodynamic integration study performed by the same group[7] using hybrid functionals[23] without nuclear quantum effects, predicted a larger value of 0.97 eV.

Here we present accurate predictions of the EA of liquid water and its surface using a combination of path-integral molecular dynamics with ab initio potentials, and MBPT calculations, thus determining a key missing property of electrons in water. We compared our results for the water surface with experiments on a-ice, thus validating our computational strategy against the only available direct measurement. We then used the same protocol to calculate the EA of the bulk, which we predict to be 0.1–0.3 eV. This value is similar to that estimated by Coe et al.[16], although we show below that the agreement is fortuitous and effectively derives from inaccurate assumptions made in ref.[16]. We also revisit several other experiments and the assumptions that led to current literature values of the EA.

## Results

**Computational model.** The structural models of liquid water were generated using path-integral molecular dynamics (MD) simulations with interatomic interactions described with the many-body MB-pol potential energy function[27–29], which has been shown to accurately predict the properties of water in gaseous and condensed phases[30]. Electronic-structure properties were then computed for MD snapshots utilizing MBPT at the $G_0W_0$ level of theory, starting from eigenvalues and wavefunctions determined from hybrid density functional calculations. The water IP was computed by extrapolation of the ($1b_1$) peak energy, similar to the procedure followed to analyze experimental data[13]; the EA was computed as the lowest unoccupied single-particle energy level of the liquid[22,31–35] (see SI). Within MBPT, the energies −IP and −EA correspond, respectively, to the valence band maximum (VBM) and conduction band minimum (CBM), which were referred to vacuum using a procedure commonly utilized to compute band offsets in semiconductors (see Supplementary Methods).

In order to validate our computational protocol, we first compared the energies computed for the occupied electronic states of water with well-established liquid jet experimental data[13,36]. Consistent with our previous studies[37,38], we found that the most accurate protocol for computing the binding energies of electrons in water is the $G_0W_0$ approximation based on dielectric-dependent hybrid (DDH) functionals[39,40]. In particular, $G_0W_0$ calculations with the range-separated hybrid (RSH) yielded a mean absolute error (MAE) with respect to measured spectra of only 0.18 eV for trajectories including nuclear quantum effects (NQE) and 0.38 eV for classical trajectories.

In addition, to assess the performance of the same level of theory for the unoccupied states of the liquid, we relied on the comparison of our results with highly-accurate quantum chemistry calculations of the EA of water clusters[41], due to the lack of experimental data for the bulk system. Specifically, we compared $G_0W_0$ electron affinities with those obtained with the coupled cluster with single, double, and perturbative triple excitations [CCSD(T)] method for a series of water hexamers, and found that overall, $G_0W_0$/RSH quasiparticle energies agree with the CCSD(T) results within 0.05 eV. This accuracy is comparable to the statistical uncertainty of our calculations.

**Energy levels of bulk water and the water surface.** Figure 1 (left-hand side) summarizes the computed VBM and CBM of a slab representing the water/vacuum interface, as obtained at the $G_0W_0$/RSH level. The computed CBM values of −0.88 eV ($G_0W_0$/RSH) and −0.79 eV ($G_0W_0$/sc-hybrid) are in agreement with the direct measurement of Stähler et al.[20], who reported a conduction band minimum of −0.8 eV at the interface of amorphous ice and a metal electrode. The agreement with experiment found here for the a-ice surface, expected to be representative of that of the liquid as well, further proves that our computational protocol accurately predicts the energy of the empty states of water.

We now turn to discussing the EA of the bulk liquid. As Fig. 1 shows, the position of the VBM of the surface and bulk water are similar (differing by less than 0.1 eV); however the position of the bulk CBM is ~0.6 eV higher than that of the surface. Our results obtained with $G_0W_0$/RSH (−0.29 eV) and $G_0W_0$/sc-hybrid (−0.17 eV) yield EA of the bulk within 0.1 and 0.3 eV, when including statistical uncertainties of our calculations (0.04–0.05 eV). This range has been obtained using MD trajectories inclusive of nuclear quantum effects; with classical trajectories and the same level of electronic structure theory, we found smaller values: 0 to 0.1 eV.

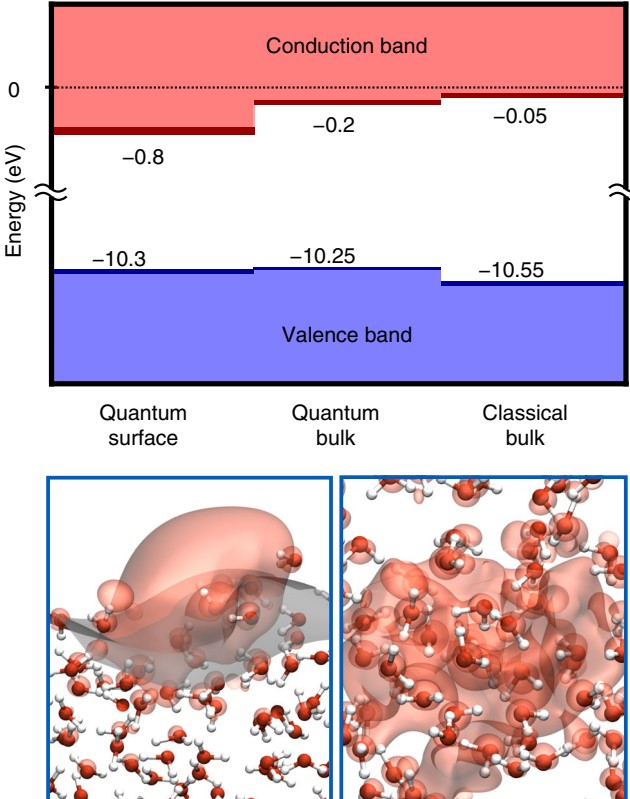

**Fig. 1** Computed electronic energy levels of liquid water. Upper panel: Positions of the valence band maxima (VBM, blue) and conduction band minima (CBM, red) of the surface of the water and bulk water computed using the classical and path integral molecular dynamics with the MB-pol potential. All values are in eV. The energy levels were computed using $G_0W_0$ starting from hybrid DFT; the range given above (thick bars) corresponds to results obtained with range-separated (RSH) and self-consistent hybrids (see text and Supplementary Tables 7–10). Lower panel: Snapshots of the surface of water (left) and of bulk liquid water (right) from path integral MD simulations, together with isoprobability contours (set at 40%) of the lowest unoccupied molecular orbital, as obtained using the RSH functional. The interface between water and vacuum (gray isosurface in the lower left panel) is pictorially represented using a constant density surface defined following the procedure reported in ref.[69]

**Nuclear quantum effects**. The results presented in Fig. 1 highlight the influence of nuclear quantum effects on the electronic structure of liquid water. In particular, the inclusion of NQE affects the VBM and CBM position by 0.3 and 0.2 eV, respectively (the energy of the VBM is increased and that of the CBM lowered) yielding a gap 0.5 eV smaller than that calculated for classical trajectories. The decrease in the gap is consistent with the presence of longer O–H bonds ($0.99 \pm 0.14$ Å) when NQE are included (to be compared with $0.97 \pm 0.05$ Å for classical trajectories): Indeed, ref.[22] showed that water models with longer O–H bonds exhibited smaller gaps. Accounting for NQE also results in a wider bond-pair peak in the distribution of maximally localized Wannier centers (Supplementary Fig. 6), consistent with larger O–H bond-length fluctuations.

Our result for the fundamental gap is in agreement with the study of Chen et al.[25] reporting a 0.5–0.7 eV reduction of the band gap of water due to nuclear quantum effects. Instead our findings are inconsistent with the density of virtual states reported in ref.[34] showing a rather large shift of the CB edge upon inclusion of NQE (of almost 1 eV). We also note that the fluctuations of the VBM and CBM observed in our quantum

trajectories are of the order of 0.4 and 0.1 eV for the bulk and 0.6 and 0.4 eV for the surface, indicating substantial fluctuations of band edges depending on the nuclear configuration, which should be taken into account when modeling chemical reactions at aqueous interfaces.

The marked difference in the EA of the bulk and the surface is reflected in the localization properties of their respective conduction band edges. In Fig. 1 we report the isosurface of the wavefunctions corresponding to the CBM of the surface and the bulk, showing the different localization properties of the two states. Consistent with previous reports[42,43], we found that the bulk unoccupied edge is delocalized over the entire supercell, while that of the slab is localized at the surface, in proximity of broken hydrogen bonds. The different bonding environment of the surface, with respect to that of the bulk, is reflected in the values of effective polarizabilities[44]: As expected from our previous work on salts[45], we found that the effective molecular polarizabilities in the slab, where more broken hydrogen bonds are present, are smaller than in the bulk. In addition the distributions of polarizabilities and dipole moments are broader in water with NQE than in the liquid treated classically (Supplementary Figs. 5 and 6).

## Discussion

Our prediction for the liquid vertical EA (0.1–0.3 eV) differs from data often quoted as experimental results in the literature, e.g., 0.74 eV from ref.[15]. However, these data are not obtained from direct measurements; rather, they are derived using both experimental and theoretical results, under specific assumptions which, as we show below, need to be revisited. By combining our theoretical prediction of the vertical EA with the most recent measurements on solvated electrons, we discuss below the construction of an energy diagram for an electron in liquid water.

The vertical (EA) and adiabatic (AEA) electron affinities of water are defined as:

$$\text{EA} = E_{\text{water}} - E_{\text{e}^-} \tag{1}$$

and

$$\text{AEA} = E_{\text{water}} - E_{\text{e}^-(\text{aq})}, \tag{2}$$

where $E_{\text{water}}$ is the energy of pristine, neutral water and $E_{\text{e}^-}$ that of the liquid with a non-solvated electron; $E_{\text{e}^-(\text{aq})}$ is the energy of the liquid with a solvated electron $\text{e}^-$ (aq). The vertical detachment energy (VDE), or energy necessary to remove an $\text{e}^-$ (aq) from water, is defined as

$$\text{VDE} = E_{\text{water}}^{\text{cavity}} - E_{\text{e}^-(\text{aq})}, \tag{3}$$

where $E_{\text{water}}^{\text{cavity}}$ refers to the energy of the neutral liquid with a defect (cavity) created by a solvated electron. The energy required to create the defect is the reorganization energy $\lambda$:

$$\lambda = E_{\text{water}}^{\text{cavity}} - E_{\text{water}}. \tag{4}$$

Finally, $\left[E_{\text{e}^-(\text{aq})}\right]^*$ is the energy of the solvated electron's first excited state with excitation energy $\mu = \left[E_{\text{e}^-(\text{aq})}\right]^* - E_{\text{e}^-(\text{aq})}$.

Figure 2 shows the electron energy diagram in liquid water constructed from our EA calculations and the most recent measurements for AEA, VDE, and $\mu$. Specifically, AEA = 1.34 eV was obtained by extrapolation of water cluster data[46]; VDE = 3.7 eV was measured by photoemission spectroscopy including corrections for surface scattering effects[47]; and $\mu = 1.73$ eV is a well-established position of the maximum in the measured optical absorption spectrum of the solvated electron[48]. The reorganization energy $\lambda$ derived from the data of Fig. 2 is 2.36 eV, larger

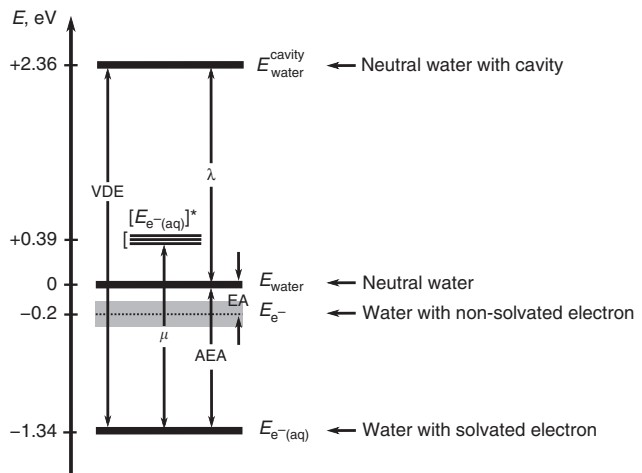

**Fig. 2** Energy diagram of an electron in water. The energies reported in this diagram are from the most recent experimental results available in the literature and the value of the vertical electron affinity (EA) is determined in this work (0.1–0.3 eV). The vertical detachment energy of the solvated electron is VDE = 3.7 eV[47] and the adiabatic electron affinity of water is AEA = 1.34 eV[46]. The difference between VDE and AEA is the water reorganization energy upon solvation of an electron: $\lambda = 2.36$ eV; note the difference of more than 1 eV with the values used in the left panel of Fig. 3. $\left[E_{e^-(aq)}\right]^*$ corresponds to the excited state of the solvated electron e[−] (aq), with excitation energy $\mu = 1.73$ eV[48]. All values are in eV

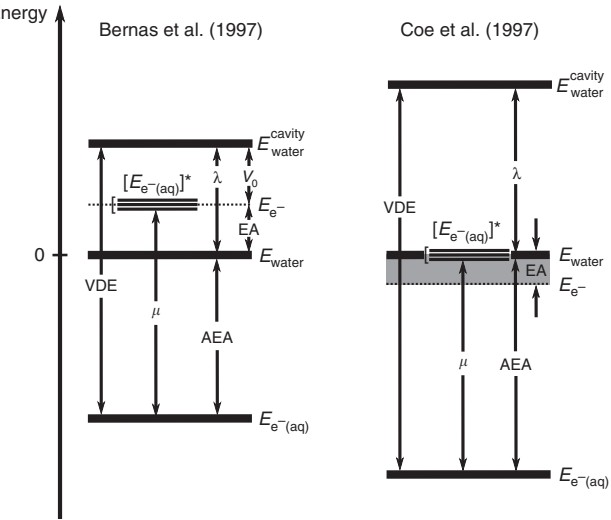

**Fig. 3** Energy diagrams of solvated electron inferred from past work. The diagrams are derived from the work of Bernas et al.[15] (left) and Coe et al.[16] (right). The notation is the same as in Fig. 2. The zero of both diagrams corresponds to the energy of pristine, neutral water, $E_{water}$. Bernas et al. used AEA = 1.47 eV[70], while Coe et al. adopted AEA = 1.72 eV. On the left, $\lambda = 1$ eV[49, 50] was taken from a theoretical study and $\mu = 1.73$ eV from optical absorption experiments[48]. On the right, VDE = 3.32 eV[55] was extrapolated from cluster data. Ref.[15] (left) assumed that the energy of pristine water with an added electron (prior to ionic relaxation) coincides with the first excited state of the solvated electron $\left[E_{e^-(aq)}\right]^*$

than the most recent computational estimates of 1.99–2.17 eV[7].

The diagram obtained here differs substantially from those obtained using the values reported by Bernas et al.[15] and by Coe et al.[16], shown on the left-hand and right-hand sides of Fig. 3, respectively. Focusing first on the results of ref.[15], the authors considered AEA = 1.47 eV, $\lambda = 1$ eV[49,50], and $\mu = 1.73$ eV[48]. This led to a VDE of 2.47 eV, which is significantly different from that reported by recent measurements (3.3–3.8 eV)[1,14,47,51–53]. This discrepancy originates from the incorrect magnitude of the reorganization energy used in ref.[15], a value obtained from classical MD simulations[49,50], which turned out to be underestimated by more than 1 eV with respect to all best estimates known today. In addition, a crucial assumption of ref.[15] was that the first excited state of the solvated electron coincides with the bottom of the conduction band of water, hence $E_{e^-} \equiv \left[E_{e^-(aq)}\right]^*$. This assumption was later disproved and is now known to be incorrect[54].

Finally, we note that Bernas et al.[15] defined the bottom of the conduction band of water from the energy difference $V_0 = E_{e^-} - E_{water}^{cavity}$, which, however, is not equal to −EA. Using the energy diagram of Fig. 3, one would obtain a negative value for the EA of liquid water: −0.26 eV.

Coe et al.[16] (right-hand-side diagram) used the values VDE = 3.32 eV and AEA = 1.72 eV, both derived from extrapolated experimental data for water clusters[55], and obtained a larger reorganization energy ($\lambda = 1.60$ eV) than Bernas et al.[15] They then computed an upper bound to the EA as (AEA − $\lambda$) by assuming that the energy difference ($E_{e^-} - E_{e^-(aq)}$) is always smaller than the reorganization energy. However, the derivation incorrectly assumed that the creation of a cavity in pure water implies an energy gain, rather than a cost, as pointed out in ref.[7]. If energy gains and costs are correctly taken into account, the only conclusion that can be drawn from the data of ref.[16] is EA ≤ 1.72 eV.

We compare our results with other computational studies, focusing on first-principles calculations. Using several water models, previous studies predicted the EA of the bulk between 0.7

and 0.9 eV at the $G_0W_0$/PBE level of theory. As shown in Supplementary Table 7, we obtained a similar result (0.99 eV) for classical MB-pol water with the same electronic structure methods. Our best prediction for the bulk water EA is smaller than that reported recently using first-principles MD trajectories with van der Waals functionals and nuclear quantum effects (0.5 eV)[25]. These calculations used self-consistent GW with approximate vertex corrections and smaller cells (32 water molecules) than those adopted in our study (64–256 molecules).

Knowledge of the water EA is key to understanding mechanisms of redox reactions in aqueous systems that involve either molecular species or solid surfaces[53,56,57], yet no direct experimental measurement is available. In this paper we reported predictions of the EA of bulk water and the surface of water, obtained entirely from first principles, by combining path-integral MD simulations with ab initio potentials, and state-of-the-art electronic structure methods based on MBPT. The accuracy of the theoretical methods used in this work was carefully checked against quantum chemistry calculations for virtual states of aqueous systems[41], and against well-established experiments for electronic binding energies of the liquid. Our results for the surface were found to be in good agreement with recent pump-probe spectroscopic measurements[20], thus further validating the accuracy of our computational framework.

We determined the EA of the liquid to be between 0.1 and 0.3 eV, much smaller than some of the accepted estimates present in the literature, but consistent with the speculations of Coe et al.[16,18] and the measurements of the electron ejection lengths in a two-photon ionization process by the Bradforth group[58,59] (The authors of refs.[58,59] determined that the average ejection length of the electrons in a two-photon ionization process is roughly constant below 9.5 eV but increases rapidly after 9.5–9.8 eV, and associated this increase with the electrons being able to access the conduction band upon excitation. Given that the ionization threshold of water is 9.9 eV[13], the energy of 9.5–9.8 eV

corresponds to an energy level positioned ~0.1–0.4 eV below vacuum, although refs.[58,59] do not explicitly mention these values for the conduction band minimum). We used our theoretical prediction of the vertical EA together with the most recent measurements for the solvated electron, to construct an energy level diagram for an electron in water, which we believe is the most accurate known to date. The diagram proposed here differs substantially from those generated using estimates reported in the literature, for which we provided a detailed, critical reassessment. We suggest that the EA of water may be determined experimentally by extending the pump-probe spectroscopic study of ref.[20] to thicker a-ice films and extrapolating the results as a function of size, building, e.g., on work reported by King et al.[21] on trapped electrons at interfaces.

Our results showed that while the IP of bulk water and the surface are very similar, their EAs differ by more than 0.6 eV, with the conduction band edge of the surface being much deeper in energy than that of the bulk. We also found that the band gap of water as well as both band edges are substantially affected by proton quantum effects and couplings with nuclear configurations, highlighting the importance of including energy level fluctuations when modeling chemical reactions in water. Work is in progress to investigate the same effects in aqueous solutions.

## Methods

**Molecular dynamics simulations.** Classical and path-integral (PI) molecular dynamics (MD) simulations were performed using the ab initio MB-pol potential and a modified version of the DL_POLY code. Bulk water was simulated in the constant volume-constant temperature (NVT) ensemble at 298 K, using a Nosé–Hoover chain thermostat[60] with the length of 4, and supercells containing 64 molecules. The equilibrium density ($\rho$) of the liquid was determined with constant pressure and temperature (NPT) simulations, corresponding to cell sizes of 12.41 $\text{Å}^3$ ($\rho = 1.001$ g cm$^{-3}$) and 12.42 $\text{Å}^3$ ($\rho = 0.999$ g cm$^{-3}$) for classical and PI molecular-dynamics simulations, respectively. Path-integral simulations were performed using 32 beads per atoms. The water surface was simulated with a slab containing 108-molecules, in a cell of dimensions $12.74 \times 12.74 \times 57.35$ $\text{Å}^3$. After equilibration, we carried out simulations for 0.5–1 ns for the bulk and the slab. In particular, we insured that the electric field across the water slab vanished within 0.0005 eV/Å. We tested size effects on the position of energy levels by performing MD simulations for 256-molecule bulk water and 216- and 384-molecule slabs. The details of the simulations used for finite size-scaling are discussed in the Supplementary Methods. We note that the choice of the system size in our MD simulations was limited by the cost of MBPT calculations on MD trajectories.

**Electronic structure calculations.** We carried out calculations of the electronic properties of water and water slabs at the density-functional theory (DFT) and $G_0W_0$ level. DFT calculations were carried out using the Quantum Espresso code[61], with a plane-wave energy cutoff of 85 Ry, and HSCV pseudopotentials[62,63]. $G_0W_0$ calculations were performed using the West code[64,65], starting from wavefunctions obtained with the PBE[66], PBE0[67], sc-hybrid[39], and RSH[40] functionals. Quasiparticle energies were determined using the same exchange-correlation potential as the one employed in the self-consistent-field DFT calculations. All of our $G_0W_0$ calculations were carried out with 1600 eigenpotentials and extrapolated using the procedure described in the Supplementary Table 5. DFT calculations were done for 128 equally-spaced snapshots along each of the bulk trajectories. $G_0W_0$ calculations for bulk water, starting from hybrid functional wavefunctions, were performed for 4 snapshots. The $G_0W_0$ results for the slabs were obtained using computed DFT values for the slab and $G_0W_0$ corrections for the bulk. Our analysis of the accuracy of this approach and error estimates are given in the Supplementary Methods.

To determine absolute energy levels from DFT and $G_0W_0$ calculations performed in plane-wave basis sets, we aligned the plane-average electrostatic potentials in the bulk water and in the slab, and referenced them to the vacuum energy level, following the procedure proposed by van de Walle and Martin[68], as explained in the Supplementary Methods.

**Data availability.** All data associated with our article are available at https://datahub.uchicago.edu/.

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

## Acknowledgements

The authors gratefully acknowledge helpful discussions with Stephen Bradforth, Nicholas Brawand, Federico Giberti, François Gygi, Ryan McAvoy, and Robert Seidel. A.P.G., M.G., and G.G. were supported by MICCoM as part of the Computational Materials Sciences Program funded by the U.S. Department of Energy (DOE), Office of Science, Basic Energy Sciences (BES), Materials Sciences and Engineering Division (5J-30161-0010A). A.P.G. was also supported by the postdoctoral fellowship from the Natural Sciences and Engineering Research Council of Canada. T.A.P. was supported by the Lawrence Fellowship. F.P. was supported by the National Science Foundation through grant CHE-1453204 and used the Extreme Science and Engineering Discovery Environment (XSEDE), which is supported by the National Science Foundation through grant ACI-1053575. Part of this work was performed under the auspices of the U.S. DOE at Lawrence Livermore National Laboratory under contract DE-AC52-07A27344. An award of computer time was provided by the INCITE program. This research used resources of the Argonne Leadership Computing Facility, which is a DOE Office of Science User Facility supported under contract DEAC02-06CH11357.

## Author contributions

A.P.G., T.A.P., F.P., and G.G. designed the research. T.A.P. and F.P. performed MB-pol simulations. A.P.G. performed most of the electronic-structure calculations and data analysis. M.G. implemented the $G_0W_0$ method and DDH functionals in the West code. A.P.G. and G.G. wrote the manuscript with contributions from F.P. and T.A.P. All authors contributed to the discussion of the results.

## Additional information

**Competing interests:** The authors declare no competing financial interests.

