## [Peer Review File · Nature Communications]

Reviewer #1 (Remarks to the Author):

This is an extremely important and potentially influential paper making a strong case for a fundamental redrawing (qualitatively as well as quantitatively) of the bulk energy-level diagram for liquid water. That is important perhaps for obvious reasons, but in particular the value of 0.74 eV for the vertical EA has been in the literature for 20 years and is suggested here to be erroneous by at least 0.44 eV. The older number is still referenced frequently and taken seriously by a certain community interested in computing redox potentials of aqueous solutes; see, e.g., recent work by Sprik et al. [J. Phys. Chem. Lett. 3, 3411 (2012)], so any adjustment of the EA potentially readjusts a lot of other numbers. Finally, the present calculations have some bearing on a controversy over several years about how (and whether) the solvated electron differs in bulk water vs. the air/water interface.

Overall I think the calculations are quite thorough and carefully done (use of a good water potential, inclusion of nuclear quantum effects, careful consideration of size effects in GW calculations, etc.) I therefore recommend publication with only a few minor modifications.

(1) It is argued that the previous estimate by Coe et al. (~ 0.1 eV) is numerically right but based on flawed reasoning, namely, an assumption that creating a cavity in water "implies an energy gain, rather than a cost". While I did not go back to read the original ref. 19, this seems bizarre, if I interpret the present writing correctly. Did Coe et al. really assume that creating a cavity in water is energetically downhill? This deserves clarification, and if that's really what was assumed, it would be very useful if the authors could provide a bit more of the rationale from ref. 19.

(2) If Fig. 1 is to be used to discuss localization (which it is, in the text), then what the authors really need to plot are isoprobability contours, each containing a consistent fraction of the wave function. Only then can one eyeball the figures side-by-side and make statements about degree of localization.

(3) This is a minor quibble about terminology. The term "vibronic coupling" comes up at least twice, and to a certain community of spectroscopists that term clearly implies breakdown of the Born-Oppenheimer approximation, such that one may no longer consider individual states but only vibronic ones, due to nuclear motion near a conical intersection. That is **not** what is meant here; rather, the authors are simply referring to the somewhat large (up to 0.6 eV, in some cases) fluctuations in the electronic energy levels as a function of nuclear dynamics. These are separate ideas, and with the former interpretation of "vibronic" being firmly established, I suggest replacing the word here, in favor of some terminology involving "fluctuations".

Reviewer #2 (Remarks to the Author):

Authors describe a new calculation of the electron affinity of water. This is an important quantity in several processes involving the chemistry of water, but yet is poorly known. Their calculated value is put into context by drawing an energy diagram of some states involving a free electron in water, using recent values from the literature. Doing so, the authors correct mistakes in other such diagrams reported in the literature.

The calculational methods are state-of-the-art, as far as I can say. The manuscript is well written, and the length adequate to the content.

The authors point out that there is no direct experimental way to measure the vertical EA of water. Critically one could therefore say that diagrams such as Fig. 2 are useful to put the calculated figures into context, but do not provide any way to check their correctness. The value of this figure and its discussion therefore is rather pedagogical.

The manuscript would be much stronger if the authors could line out an experiment or a protocol how these numbers could be deduced in an experiment, or if they could provide more concrete information where they matter. What if the value were different, say 0.5 eV? Where would it matter?

I do not understand in which way Ref.s 73, 74 support the findings presented here. If the authors think so, please expand.

I suggest to state the definition of the water EA (eq.s (1), (2), and possibly the subsequent equations) earlier in the text.

I suggest to state the numerical value of the EA found in this work in the abstract.

I suggest to use colour in Fig. 2 to distinguish between values calculated in this work, values taken from literature and derived values.

The following article seems as relevant as several other works cited by the authors: DOI: [10.1021/acs.jpca.5b04721](https://doi.org/10.1021/acs.jpca.5b04721)

We thank both Reviewers for the positive comments about our work and for their constructive criticisms. Below are point-by-point answers to the Reviewers’ questions.

Responses to Reviewer 1

(1) It is argued that the previous estimate by Coe et al. (~ 0.1 eV) is numerically right but based on flawed reasoning, namely, an assumption that creating a cavity in water “implies an energy gain, rather than a cost”. While I did not go back to read the original ref. 19, this seems bizarre, if I interpret the present writing correctly. Did Coe et al. really assume that creating a cavity in water is energetically downhill? This deserves clarification, and if that’s really what was assumed, it would be very useful if the authors could provide a bit more of the rationale from ref. 19.

The assumption that creating a cavity in water is downhill is implicit in the derivation of V_0 reported by Coe et al. Specifically, referring to Fig. 3 of Ref. 19, the authors pointed out that the difference between the level of the solvated electron and the conduction band minimum $\Delta E_{\text{defect}}(e^-)$ (which corresponds to $[E_{e^-(\text{aq})} - E_{e^-}]$, following the notation used in our paper) is the sum of the reorganization energy, RE (λ) and charge stabilization energy. It is then stated that “The RE associated with the hydrated electron represents a minimum magnitude for $\Delta E_{\text{defect}}(e^-)$, so $-\Delta E_{\text{defect}}(e^-) \geq \text{RE}$.” This implies that the magnitude of $\Delta E_{\text{defect}}(e^-)$ is equal to the *sum of magnitudes* of RE and the charge stabilization energy, which can only be the case if RE and the charge stabilization energy have the same sign. However, these quantities have different signs, as the RE represents an energy cost, while the charge stabilization represents an energy gain. Using this incorrect logic, Coe et al. derived the following expression for the lower bound of the conduction band minimum: $V_0 \geq \lambda - \text{AEA} = 1.60 \text{ eV} - 1.72 \text{ eV} = -0.12 \text{ eV}$. This result—although it stems from an incorrect logic—is not incorrect by itself; it is just not useful to predict the value of V_0 .

(2) If Fig. 1 is to be used to discuss localization (which it is, in the text), then what the authors really need to plot are isoprobability contours, each containing a consistent fraction of the wave function. Only then can one eyeball the figures side-by-side and make statements about degree of localization.

We thank the Reviewer for this useful suggestion. We modified the isosurfaces in both panels of Fig. 1 to include exactly 40% of the LUMO density, and revised the figure caption.

*(3) This is a minor quibble about terminology. The term “vibronic coupling” comes up at least twice, and to a certain community of spectroscopists that term clearly implies breakdown of the Born-Oppenheimer approximation, such that one may no longer consider individual states but only vibronic ones, due to nuclear motion near a conical intersection. That is **not** what is meant here; rather, the authors are simply referring to*

the somewhat large (up to 0.6 eV, in some cases) fluctuations in the electronic energy levels as a function of nuclear dynamics. These are separate ideas, and with the former interpretation of “vibronic” being firmly established, I suggest replacing the word here, in favor of some terminology involving “fluctuations”.

We agree with the Reviewer that the use of “vibronic coupling” could be potentially misleading, especially for the quantum chemistry community. We revised the language of the manuscript and used the words “fluctuations of band edges”.

Responses to Reviewer 2

The manuscript would be much stronger if the authors could line out an experiment or a protocol how these numbers could be deduced in an experiment, or if they could provide more concrete information where they matter. What if the value were different, say 0.5 eV? Where would it matter?

We included a statement into the manuscript about suggested experiments that could be performed to probe the electron affinity of bulk water: “We suggest that the electron affinity of water may be determined experimentally by extending the pump-probe spectroscopic study of Ref. 1 to thicker a-ice films and extrapolating the results as a function of size, building, e.g. on work reported by King et al.² on trapped electrons at interfaces.”

The knowledge of the exact position of the conduction band minimum of water would be important, e.g., for determining the mechanisms of reactions involving solvated electrons.^{3,4} As an example, the electrode potential of the $\text{N}_2 + \text{e}^- \longrightarrow \text{N}_2^-(\text{aq})$ half-reaction is -4.2 eV with respect to the standard hydrogen electrode (SHE), so this reaction could be driven by an electron at the bottom of the conduction band as determined in our work (-4.34 to -4.14 eV) but not as determined by the value of Bernas et al.⁵ Based on the results of our work, we would suggest that this reaction should be considered in the search for the mechanism of the process described in Ref. 3. We modified the first sentence of the Conclusions as “Knowledge of the water electron affinity is key to understanding mechanisms of redox reactions in aqueous systems that involve either molecular species^{3,4,6,7} or solid surfaces⁶...”

I do not understand in which way Refs. 73, 74 support the findings presented here. If the authors think so, please expand.

The authors of Refs. 73 and 74 determined that the average ejection length of the electrons in a two-photon ionization process is roughly constant below 9.5 eV but increases rapidly after 9.5–9.8 eV. They associated this increase in the ejection length with the electrons being able to access the conduction band upon excitation. The ionization threshold of water (corresponding to the

electrons exiting the liquid into vacuum with a zero kinetic energy) is 9.9 eV.⁸ The energy of 9.5–9.8 eV corresponds to an energy level positioned ~ 0.1 – 0.4 eV below vacuum, although Refs. 73 and 74 do not explicitly mention this value for the conduction band minimum. Ref. 73 states that “The close agreement of the energy range over which the electron ejection distance in the liquid phase rises steeply with the range over which the liquid photoelectron spectrum has a large vertical transition probability to vacuum is in agreement with the assignment of Coe et al. that V_0 is very small for liquid water.” Furthermore, we discussed our work with Stephen Bradforth, the senior author of both Refs. 73 and 74, and he confirmed our understanding of their findings, and agreement with our own calculations (0.1–0.3 eV).

To make this point more explicit, we modified the sentence comparing our results with those of Elles et al. as follows: “We determined the EA of the liquid to be between 0.1 and 0.3 eV, much smaller than some of the accepted estimates present in the literature, but consistent with the speculations of Coe et al.^{9–11} and the measurements of the electron ejection lengths in a two-photon ionization process by the Bradforth group.^{12,13}” We also added a footnote that reads: “The authors of Refs. 12 and 13 determined that the average ejection length of the electrons in a two-photon ionization process is roughly constant below 9.5 eV but increases rapidly after 9.5–9.8 eV, and associated this increase with the electrons being able to access the conduction band upon excitation. Given that the ionization threshold of water is 9.9 eV,⁸ the energy of 9.5–9.8 eV corresponds to an energy level positioned ~ 0.1 – 0.4 eV below vacuum, although Refs. 12 and 13 do not explicitly mention these values for the conduction band minimum.”

I suggest to state the definition of the water EA (eq.s (1), (2), and possibly the subsequent equations) earlier in the text.

We feel that discussing the definition of the electron affinity as the difference of total energies is best done in the Discussion section because we never use Eqs. (1) and (2) to actually compute the electron affinity of the liquid water. (Instead, we estimate the EA as a quasiparticle energy of the first empty state.) These equations are shown only when discussing our results in the context of past work, and only in the proximity of Figs. 2 and 3 introducing the energy levels that are part of Eqs. (1) and (2). We believe this description fits the logic of our narration best.

However, we would like to consider the Reviewer’s comment in a broader sense, in that more clarity is needed in discussing this important quantity. Hence, we added a statement “the liquid electron affinity, or the energy gain due to the injection of an electron into the liquid” when introducing the electron affinity in the introduction.

I suggest to state the numerical value of the EA found in this work in the abstract.

We agree with this suggestion of the Reviewer and revised the language of the abstract accordingly.

I suggest to use colour in Fig. 2 to distinguish between values calculated in this work, values taken from literature and derived values.

We tried to use color coding but we felt that it would complicate the analysis of Fig. 2 and its comparison with Fig. 3. For example, we would need to use the same color to denote our G_0W_0 calculations of the electron affinity of water and the classical MD calculation of the reorganization energy in Refs. 60 and 61 (as we would have to choose one color for “theoretical results”), although the two calculations were conducted at very different levels of theory and their accuracy is rather different. Also, some of the values, such as VDE and AEA in Fig. 3, were obtained by extrapolating cluster data; even though these data are coming from experimental measurements, the final value is the result of an extrapolation, not a directly measured quantity, so we cannot label that quantity as “measured”. Finally, the purpose of the diagrams of Figs. 2 and 3 is to help explain the assumptions behind the estimation of the electron affinity of water reported by Bernas et al.⁵ and Coe et al.⁹

*The following article seems as relevant as several other works cited by the authors:
DOI: 10.1021/acs.jpca.5b04721*

We thank the Reviewer for this recommendation; we have cited the paper in the Introduction, along with several other relevant papers of Jungwirth and co-workers, and Boero et al., which we originally did not include.

References

- [1] J. Stähler, J.-C. Deinert, D. Wegkamp, S. Hagen, and M. Wolf, “Real-Time Measurement of the Vertical Binding Energy during the Birth of a Solvated Electron”, *J. Am. Chem. Soc.* **137**, 3520 (2015).
- [2] S. B. King, D. Wegkamp, C. Richter, M. Wolf, and J. Stähler, “Trapped Electrons at the Amorphous Solid Water/Vacuum Interface as Possible Reactants in a Water Splitting Reaction”, *J. Phys. Chem. C* **121**, 7379 (2017).
- [3] D. Zhu, L. Zhang, R. E. Ruther, and R. J. Hamers, “Photo-illuminated diamond as a solid-state source of solvated electrons in water for nitrogen reduction”, *Nat. Mater.* **12**, 836 (2013).
- [4] L. Zhang, D. Zhu, G. M. Nathanson, and R. J. Hamers, “Selective Photoelectrochemical Reduction of Aqueous CO_2 to CO by Solvated Electrons”, *Angew. Chem. Int. Ed.* **53**, 9746 (2014).
- [5] A. Bernas, C. Ferradini, and J.-P. Jay-Gerin, “On the electronic structure of liquid water: Facts and reflections”, *Chem. Phys.* **222**, 151 (1997).
- [6] J. R. Christianson, D. Zhu, R. J. Hamers, and J. R. Schmidt, “Mechanism of N_2 Reduction to NH_3 by Aqueous Solvated Electrons”, *J. Phys. Chem. B* **118**, 195 (2014).
- [7] K. R. Siefermann, Y. Liu, E. Lugovoy, O. Link, M. Faubel, U. Buck, B. Winter, and B. Abel, “Binding energies, lifetimes and implications of bulk and interface solvated electrons in water”, *Nat. Chem.* **2**, 274 (2010).

- [8] B. Winter, R. Weber, W. Widdra, M. Dittmar, M. Faubel, and I. V. Hertel, “Full valence band photoemission from liquid water using EUV synchrotron radiation”, *J. Phys. Chem. A* **108**, 2625 (2004).
- [9] J. V. Coe, A. D. Earhart, M. H. Cohen, G. J. Hoffman, H. W. Sarkas, and K. H. Bowen, “Using Cluster Studies to Approach the Electronic Structure of Bulk Water: Reassessing the Vacuum Level, Conduction Band Edge, and Band Gap of Water”, *J. Chem. Phys.* **107**, 6023 (1997).
- [10] J. V. Coe, “Fundamental properties of bulk water from cluster ion data”, *Int. Rev. Phys. Chem.* **20**, 33 (2001).
- [11] J. V. Coe, S. M. Williams, and K. H. Bowen, “Photoelectron spectra of hydrated electron clusters vs. cluster size: connecting to bulk”, *Int. Rev. Phys. Chem.* **27**, 27 (2008).
- [12] C. G. Elles, A. E. Jailaubekov, R. A. Crowell, and S. E. Bradforth, “Excitation-energy dependence of the mechanism for two-photon ionization of liquid H₂O and D₂O from 8.3 to 12.4 eV”, *J. Chem. Phys.* **125**, 044515 (2006).
- [13] C. G. Elles, C. A. Rivera, Y. Zhang, P. A. Pieniazek, and S. E. Bradforth, “Electronic structure of liquid water from polarization-dependent two-photon absorption spectroscopy”, *J. Chem. Phys.* **130**, 084501 (2009).

Reviewer #1 (Remarks to the Author):

I was satisfied with the original manuscript up to a few clarifications, which the authors have made here. This is important work and I strongly support publication.

Reviewer #2 (Remarks to the Author):

Satisfactory revision, proceed to publication.